# Sequential Neural Processes

**Gautam Singh**[*]
Rutgers University
singh.gautam@rutgers.edu

**Jaesik Yoon**[*]
SAP
jaesik.yoon01@sap.com

**Youngsung Son**
ETRI
ysson@etri.re.kr

**Sungjin Ahn**
Rutgers University
sungjin.ahn@rutgers.edu

## Abstract

Neural Processes combine the strengths of neural networks and Gaussian processes to achieve both flexible learning and fast prediction in stochastic processes. However, a large class of problems comprises underlying temporal dependency structures in a sequence of stochastic processes that Neural Processes (NP) do not explicitly consider. In this paper, we propose Sequential Neural Processes (SNP) which incorporates a temporal state-transition model of stochastic processes and thus extends its modeling capabilities to dynamic stochastic processes. In applying SNP to dynamic 3D scene modeling, we introduce the Temporal Generative Query Networks. To our knowledge, this is the first 4D model that can deal with the temporal dynamics of 3D scenes. In experiments, we evaluate the proposed methods in dynamic (non-stationary) regression and 4D scene inference and rendering.

## 1 Introduction

Neural networks consume all training data and computation through a costly training phase to engrave a single function into its weights. While this makes us entertain fast prediction on the learned function, under this rigid regime changing the target function means costly retraining of the network. This lack of flexibility thus plays as a major obstacle in tasks such as meta-learning and continual learning where the function needs to be changed over time or on-demand. Gaussian processes (GP) do not suffer from this problem. Conditioning on observations, it directly performs inference on the target stochastic process. Consequently, Gaussian processes show the opposite properties to neural networks: it is flexible in making predictions because of its non-parametric nature, but this flexibility comes at a cost of having slow prediction. GPs can also capture the uncertainty on the estimated function.

Neural Processes (NP) (Garnelo et al., 2018b) are a new class of methods that combine the strengths of both worlds. By taking the meta-learning framework, Neural Processes *learn to learn* a stochastic process quickly from observations while experiencing multiple tasks of stochastic process modeling. Thus, in Neural Processes, unlike typical neural networks, learning a function is fast and uncertainty-aware while, unlike Gaussian processes, prediction at test time is still efficient.

An important aspect for which Neural Processes can be extended is that in many cases, certain temporal dynamics underlies in a sequence of stochastic processes. This covers a broad range of problems from learning RL agents being exposed to increasingly more challenging tasks to modeling dynamic 3D scenes. For instance, Eslami et al. (2018) proposed a variant of Neural Processes, called the Generative Query Networks (GQN), to learn representation and rendering of 3D scenes. Although this was successful in modeling static scenes like fixed objects in a room, we argue that to handle

---

[*]Equal contribution

more general cases such as dynamic scenes where objects can move or interact over time, we need to explicitly incorporate a temporal transition model into Neural Processes.

In this paper, we introduce Sequential Neural Processes (SNP) to incorporate the temporal state-transition model into Neural Processes. The proposed model extends the potential of Neural Processes from modeling a stochastic process to modeling a dynamically changing sequence of stochastic processes. That is, SNP can model a (sequential) stochastic process of stochastic processes. We also propose to apply SNP for dynamic 3D scene modeling by developing the Temporal Generative Query Networks (TGQN). In experiments, we show that TGQN outperforms GQN in terms of capturing transition stochasticity, generation quality, generalization to time-horizons longer than those used during training.

Our main contributions are: We introduce Sequential Neural Processes (SNP), a meta-transfer learning framework for a sequence of stochastic processes. We realize SNP for dynamic 3D scene inference by introducing Temporal Generative Query Networks (TGQN). To our knowledge, this is the first 4D generative model that models dynamic 3D scenes. We describe the training challenge of *transition-collapse* unique to SNP modeling and resolve it by introducing the *posterior-dropout* ELBO. We demonstrate the generalization capability of TGQN beyond the sequence lengths used during training. We also demonstrate meta-transfer learning and improved generation quality in contrast to Consistent Generative Query Networks (Kumar et al., 2018) gained from the decoupling of temporal dynamics from the scene representations.

## 2 Background

In this section, we introduce notations and foundational concepts that underlie the design of our proposed model as well as motivating applications.

**Neural Processes.** Neural Processes (NP) model a stochastic process mapping an input $x \in \mathbb{R}^{d_x}$ to a random variable $Y \in \mathbb{R}^{d_y}$. In particular, an NP is defined as a conditional latent variable model where a set of *context* observations $C = (X_C, Y_C) = \{(x_i, y_i)\}_{i \in \mathcal{I}(C)}$ is given to model a conditional prior on the latent variable $P(z|C)$, and the *target* observations $D = (X, Y) = \{(x_i, y_i)\}_{i \in \mathcal{I}(D)}$ are modeled by the observation model $p(y_i|x_i, z)$. Here, $\mathcal{I}(\mathcal{S})$ stands for the set of data-point indices in a dataset $\mathcal{S}$. This generative process can be written as follows:

$$P(Y|X,C) = \int P(Y|X,z)P(z|C)\mathrm{d}z \tag{1}$$

where $P(Y|X,z) = \prod_{i \in \mathcal{I}(D)} P(y_i|x_i, z)$. The dataset $\{(C_i, D_i)\}_{i \in \mathcal{I}_{\text{dataset}}}$ as a whole contains multiple pairs of context and target sets. Each such pair $(C, D)$ is associated with its own stochastic process from which its observations are drawn. Therefore NP flexibly models multiple tasks i.e. stochastic processes and this results in a meta-learning framework. It is sometimes useful to condition the observation model directly on the context $C$ as well, i.e., $p(y_i|x_i, s_C, z)$ where $s_C = f_s(C)$ with $f_s$ a deterministic context encoder invariant to the ordering of the contexts. A similar encoder is also used for the conditional prior giving $p(z|C) = p(z|r_C)$ with $f_r(C)$. In this case, the observation model uses the context in two ways: a noisy latent path via $z$ and a deterministic path via $s_C$.

The design principle underlying this modeling is to infer the target stochastic process from contexts in such a way that sampling $z$ from $P(z|C)$ corresponds to a function which is a realization of a stochastic process. Because the true posterior is intractable, the model is trained via variational approximation which gives the following evidence lower bound (ELBO) objective:

$$\log P_\theta(Y|X,C) \geq \mathbb{E}_{Q_\phi(z|C,D)}\left[\log P_\theta(Y|X,z)\right] - \mathbb{KL}(Q_\phi(z|C,D) \parallel P_\theta(z|C)). \tag{2}$$

The ELBO is optimized using the reparameterization trick (Kingma & Welling, 2013).

**Generative Query Networks.** The Generative Query Network (GQN) can be seen as an application of the Neural Processes specifically geared towards 3D scene inference and rendering. In GQN, query $x$ corresponds to a camera viewpoint in a 3D space, and output $y$ is an image taken from the camera viewpoint. Thus, the problem in GQN is cast as: given a context set of viewpoint-image pairs, (i) to infer the representation of the full 3D space and then (ii) to generate an observation image corresponding to a given query viewpoint.

In the original GQN, the prior is conditioned also on the query viewpoint in addition to the context, i.e., $P(z|x, r_C)$, and thus results in inconsistent samples across different viewpoints when modeling

uncertainty in the scene. The Consistent GQN (Kumar et al., 2018) (CGQN) resolved this by removing the dependency on the query viewpoint from the prior. This resulted in $z$ to be a summary of a full 3D scene independent of the query viewpoint. Hence, it is consistent across viewpoints and more similar to the original Neural Processes. For the remainder of the paper, we use the abbreviation *GQN* for CGQN unless stated otherwise.

For inferring representations of 3D scenes, a more complex modeling of latents is needed. For this, GQN uses ConvDRAW (Gregor et al., 2016), an auto-regressive density estimator performing $P(z|C) = \prod_{l=1}^{L} P(z^l | z^{<l}, r_C)$ where $L$ is the number of auto-regressive rollout steps and $r_C$ is a pooled context representations $\sum_{i \in \mathcal{I}(C)} f_r(x_i, y_i)$ with $f_r$ an encoding network for context.

**State-Space Models.** State-space models (SSMs) have been one of the most popular models in modeling sequences and dynamical systems. The model is specified by a state transition model $P(z_t|z_{t-1})$ that is sometimes also conditioned on an action $a_{t-1}$, and an observation model $P(y_t|z_t)$ that specifies the distribution of the (partial and noisy) observation from the latent state. Although SSMs have good properties like modularity and interpretability due to the Markovian assumption, the closed-form solution is only available for simple cases like the linear Gaussian SSMs. Therefore, in many applications, SSMs show difficulties in capturing nonlinear non-Markovian long-term dependencies (Auger-Méthé et al., 2016). To resolve this problem, RNNs have been combined with SSMs (Zheng et al., 2017). In particular, the Recurrent State-Space Model (RSSM) (Hafner et al., 2018) maintains both a deterministic RNN state $h_t$ and a stochastic latent state $z_t$ that are updated as follows:

$$h_t = f_{\text{RNN}}(h_{t-1}, z_{t-1}), \qquad z_t \sim p(z_t|h_t), \qquad y_t \sim p(y_t|h_t, z_t). \qquad (3)$$

Thus, in RSSM, the state transition is dependent on all the past latents $z_{<t}$ and thus non-Markovian.

## 3 Sequential Neural Processes

In this section, we describe the proposed *Sequential Neural Processes* which combines the merits of SSMs and Neural Processes and thus enabling it to model temporally-changing stochastic processes.

### 3.1 Generative Process

Consider a sequence of stochastic processes $\mathcal{P}_1, \ldots, \mathcal{P}_T$. At each time-step $t \in [1, T]$, for a true stochastic process $\mathcal{P}_t$, consider drawing a set of context observations $C_t = \{(x_i^t, y_i^t)\}_{i \in \mathcal{I}(C_t)}$ where $\mathcal{I}(C_t)$ are the indices of the observations. Size of this context set may differ over time or it may even be empty. The $C_t$ are provided to the model at their respective time-steps and we want SNP to model $\mathcal{P}_t$ as a distribution over a latent variable $z_t$, as modeled in NP.

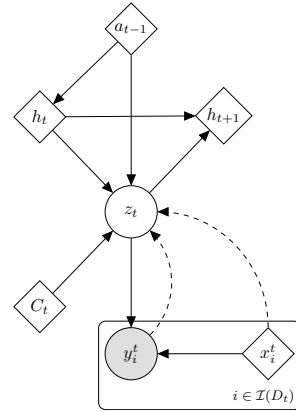

While NP models $z_t$ only using $C_t$ i.e., $P(z_t|C_t)$, in SNP we want to utilize the underlying temporal structure which governs the temporal change in the true stochastic processes $\mathcal{P}_{t-1} \rightarrow \mathcal{P}_t$. We achieve this by providing the latents of the past stochastic processes $z_{<t}$ to the distribution of the current $z_t$ resulting in $P(z_t|z_{<t}, C_t)$. Here $z_{<t}$ may be represented as an RNN encoding. The sampled latent $z_t$ is then used to model the target observation set $D_t = (X_t, Y_t)$ through $P(Y_t|X_t, z_t)$. Like $C_t$, we assume that $D_t$ is also drawn from the true process $\mathcal{P}_t$. With an abuse of notation, we use $C$, $D$, $X$, and $Y$ to bundle together the $C_t$, $D_t$, $X_t$, and $Y_t$ for all time-steps $t \in [1, T]$, e.g., $C = (C_1, \ldots, C_T)$. With these notations, the generative process of SNP is as follows:

**Figure 1:** Generative and inference (shown as dashed edges) models in TGQN.

$$P(Y, Z|X, C) = \prod_{t=1}^{T} P(Y_t|X_t, z_t) P(z_t|z_{<t}, C_t) \qquad (4)$$

where $P(Y_t|X_t, z_t) = \prod_{i \in \mathcal{I}(D_t)} P(y_i^t|x_i^t, z_t)$ and $z_0 = \texttt{null}$. The transition can also be conditioned on an action $a_{t-1}$, but we omit this throughout the paper for brevity.

Although we use the RSSM version of SNP in Eqn. (4) where the transition depends on all the past $z_{<t}$, what we propose is a generic SNP class of models that is compatible with a wide range of temporal transition models including the traditional state-space model (Krishnan et al., 2017) as long as the latents do not access the previous contexts $\mathcal{C}_{<t}$ directly.

Some of the properties of SNPs are as follows: (i) SNPs can be seen as a **generalization of NPs** in two ways. First, if $T = 1$, an SNP equals an NP. Second, if $D_t$ is empty for all $t < T$ and non-empty when $t = T$, SNP becomes an NP which uses the state transition as the (stochastic) context aggregator instead of the standard sum encoding. It then becomes an order sensitive encoding that can in practice be dealt with the order-shuffling on the contexts $\{C_t\}$. (ii) SNPs are a **meta-transfer learning** method. Consider, for example, a game-playing agent which, after clearing up the current stage, levels up to the next stage where more and faster enemies are placed than the previous stage. With SNP, the agent can not only meta-update the policy *with only a few observations* $C_t$ from the new stage, but it can also *transfer* the general trend from the past, namely, that there will be more and faster enemies in the future stages. As such, we can consider SNP to be a model combining temporal transfer-learning via $z_t$ and meta-learning via $C_t$.

### 3.2 Learning and Inference

Because a closed-form solution for learning and inference is not available for general non-linear transition and observation models, we train the model via variational approximation. For this, we approximate the true posterior with the following temporal auto-regressive factorization

$$P(Z|C, D) \approx \prod_{t=1}^{T} Q_\phi(z_t|z_{<t}, C, D) \tag{5}$$

with $z_0 = \texttt{null}$. Chung et al. (2015); Fraccaro et al. (2016); Krishnan et al. (2017); Hafner et al. (2018) provide various implementation options for the above approximation based on RNNs (forward or bi-directional) and the reparameterization-trick used. In the next section, we introduce a particular implementation of the above approximate posterior for an application to dynamic 3D-scene modeling.

With this approximate posterior, we train the model using the following evidence lower bound (ELBO): $\log P(Y|X, C) \geq \mathcal{L}_{\text{SNP}}(\theta, \phi) =$

$$\sum_{t=1}^{T} \mathbb{E}_{Q_\phi(z_t|\mathcal{V})} [\log P_\theta(Y_t|X_t, z_t)] - \mathbb{E}_{Q_\phi(z_{<t}|\mathcal{V})} [\mathbb{KL}(Q_\phi(z_t|z_{<t}, \mathcal{V}) \parallel P_\theta(z_t|z_{<t}, C_t))] \tag{6}$$

where $\mathcal{V} = (C, D)$ and $\log P_\theta(Y_t|X_t, z_t) = \sum_{i \in \mathcal{I}(D)} \log P_\theta(y_i^t|x_i^t, z_t)$. We use the reparameterization trick to compute the gradient of the objective. For the derivation of Eqn. (6), see Appendix B.1.

### 3.3 Temporal Generative Query Networks

Consider a room placed with an object. An agent can control the object by applying some actions such as translation or rotation. For such setups, whenever an action is applied, the scene changes and thus the viewpoint-to-image mapping of GQN learned in the past become stale because the same viewpoint now maps to a different image altogether. Although the new scene can be learned again from scratch using new context from the new scene, an ideal model would also be able to transfer the past knowledge such as object colors as well as utilizing the action to update its belief about the new scene. With a successful transfer, the model would adapt to the new scene with only small or no context from the new scene.

To develop this model, we propose applying SNP to extend GQN into Temporal GQN (TGQN) for modeling complex dynamic 3D scenes. In this setting, at time $t$, $C_t$ becomes the camera observations, $a_t$ the action provided to the scene objects, $z_t$ a representation of the full 3D scene, $X_t$ the camera viewpoints and $Y_t$ the images. TGQN draws upon the GQN implementation in multiple ways. We encode raw image observations and viewpoints into $C_t$ using the same encoder network and use a DRAW-like recurrent image renderer. Unlike GQN, to capture the transitions, we introduce the Temporal-ConvDRAW (T-ConvDRAW) where we condition $z_t^l$ on the past $z_{<t}$ via a concatenation of $(C_t, h_t, a_t)$. That is, $P(z_t|z_{<t}, C_t) = \prod_{l=1}^{L} P(z_t^l|z_t^{<l}, z_{<t}, C_t)$. Taking an RSSM approach (Hafner et al., 2018), $h_t$ is transitioned using a ConvLSTM (Xingjian et al., 2015). (See Fig. 1). In inference, to realize the distribution in Equation (5), $C_t \cup D_t$ is provided like in GQN (see Appendix C.2).

## 3.4 Posterior Dropout for Mitigating Transition Collapse

A novel part of SNP model is the use of the state transition $P(z_t|z_{<t}, C_t)$ which is not only conditioned on the past latents $z_{<t}$ but also on the context $C_t$. While this makes our model perform the meta-transfer learning, we found that it creates a tendency to ignore the context $C_t$ in the transition model. It seems that the problem lies in the KL term in Eqn. (6) which drives the training of the transition $p_\theta(z_t|z_{<t}, C_t)$. We note that the two distributions $q_\phi$ and $p_\theta$ are conditioned on the previous latents $z_{<t}$ which are sampled by providing all the available information $C$ and $D$. This produces a rich posterior with low uncertainty that makes good reconstructions via the decoder. While this is desirable modeling in general, we found that in practice it can make the KL collapse as the transition relies more on $z_{<t}$ while ignoring $C_t$.

This is a similar but not the same problem as the posterior collapsing (Bowman et al., 2015) because in our case the cause of the collapse is not an expressive decoder (e.g., auto-regressive), but a conditional prior which is already provided rich information about the sequence of tasks from one path via $z_{<t}$ and thus open a possibility to ignore the other path $C_t$. We call this the *transition collapse* problem.

To resolve this, we need a way to (i) limit the information available in $z_{<t}$ to incentivize the use of $C_t$ information when available while (ii) maintaining the high quality of the reconstructions. We introduce the *posterior-dropout* ELBO where we randomly choose a subset of time-steps $\mathcal{T} \subseteq [1, T]$. For these time-steps, the $z_t$ are sampled using the prior transition $p_\theta$. For the remaining time-steps in $\bar{\mathcal{T}} \equiv [1, T] \setminus \mathcal{T}$, the $z_t$ are sampled using the posterior transition $q_\phi$. This leads to the following approximate posterior:

$$\tilde{Q}(Z) = \prod_{t \in \mathcal{T}} P_\theta(z_t|z_{<t}, C_t) \prod_{t \in \bar{\mathcal{T}}} Q_\phi(z_t|z_{<t}, C, D) \tag{7}$$

Such a posterior limits the information contained in the past latents $z_{<t}$ and encourages $p_\theta$ to use the context $C_t$ for reducing the KL term. Furthermore, we reconstruct images only for time-steps $t \in \bar{\mathcal{T}}$ using latents sampled from $q_\phi$. This is because reconstructing the observations at those time-steps that use prior transitions does not satisfy the principle of auto-encoding, i.e., it then tries to reconstruct an observation that is not provided to the encoder and, not surprisingly, would result in blurry reconstructions and poorly disentangled latent space. Therefore, the posterior-dropout ELBO becomes: $\mathbb{E}_{\bar{\mathcal{T}}} \log P(Y_{\bar{\mathcal{T}}}|X, C) \geq \mathcal{L}_{\text{PD}}(\theta, \phi) =$

$$\mathbb{E}_{\bar{\mathcal{T}}} \left[ \mathbb{E}_{Z \sim \tilde{\mathcal{Q}}} \left[ \sum_{t \in \bar{\mathcal{T}}} \left[ \log P_\theta(Y_t|X_t, z_t) - \mathbb{KL}\left( Q_\phi(z_t|z_{<t}, C, D) \parallel P_\theta(z_t|z_{<t}, C_t) \right) \right] \right] \right] \tag{8}$$

Combining (6) and (8), we take the complete maximization objective as $\mathcal{L}_{\text{SNP}} + \alpha \mathcal{L}_{\text{PD}}$ with $\alpha$ an optional hyper-parameter. In experiments, we simply set $\alpha = 0$ at the start of the training and set $\alpha = 1$ when the reconstruction loss had saturated (see Appendix C.2.5). For derivation of Eqn. (8), see Appendix B.2.

## 4 Related Works

Modeling flexible stochastic processes with neural networks has seen significant interest in recent times catalyzed by its close connection to meta-learning. Conditional Neural Processes (CNP) (Garnelo et al., 2018a) is a precursor to Neural Processes (Garnelo et al., 2018b) which models the stochastic process *without* an explicit global latent. Without it, the sampled outputs at different query inputs are uncorrelated given the context. This is addressed by NP by introducing an explicit latent path. A discussion on NP, GQN (Eslami et al., 2018) and CGQN (Kumar et al., 2018) has been presented in Sec. 2. To improve the NP modeling further, one line of work pursues the problem of under-fitting of the meta-learned function on the context. To resolve this, attention on the relevant context points at query time is shown to be beneficial in ANP (Kim et al., 2019). Rosenbaum et al. (2018) apply GQN to more complex 3D maps (such as in Minecraft) by performing patch-wise attention on the context images.

In the domain of SSMs, Deep Kalman Filters (Krishnan et al., 2017) and DVBF (Karl et al., 2016) consist of Markovian state transition models for the hidden latents and an emission model for the observations. But instead of a Markovian latent structure, VRNN (Chung et al., 2015) and SRNN

(Fraccaro et al., 2016) introduce skip-connections to the past latents making roll-out auto-regressive. Zheng et al. (2017) and Hafner et al. (2018) propose Recurrent State-Space Models which also takes advantage of the RNNs to model long-term non-linear dependencies. Other variants and inference approximations have been explored by Buesing et al. (2018), Fraccaro et al. (2017), Eleftheriadis et al. (2017), Goyal et al. (2017) and Krishnan et al. (2017). To further model the long-term nonlinear dependencies, Gemici et al. (2017) and Fraccaro et al. (2018) attach a memory to the transition models. Mitigating transition-collapse through posterior-dropout broadly tries to bridge the gap between what the transition model sees during training and the test time. This intuition is related to *scheduled sampling* introduced by Bengio et al. (2015) which mitigates the teacher-forcing problem.

## 5 Experiments

We evaluate SNP on a toy regression task, and 2D and 3D scene modeling tasks. We use NP and CGQN as the baselines. We note that these baselines, unlike our model, directly access all the context data points observed in the past at every time-step of an episode and thus result in a strong baseline.

### 5.1 Regression

We generate a dataset consisting of sequences of functions. Each function is drawn from a Gaussian process with squared-exponential kernels. For temporal dynamics between consecutive functions in the sequence, we gradually change the kernel hyper-parameters with an update function and add a small Gaussian noise for stochasticity. For more details on the data generation, see Appendix D.1.

We explore three sub-tasks with different context regimes. In task (a), we are interested in how the transition model generalizes over the time steps. Therefore, we provide context points only in the first 10 time-steps out of 20. In task (b), we provide the context intermittently on randomly chosen 10 time steps out of 20. Our goal is to see how the model incorporates the new context information and updates its belief about the time-evolving function. In (a) and (b), the number of revealed points are randomly picked between 5 and 50 for each time-step chosen for showing the context. On the contrary, in task (c), we shrink this context size to 1 and provide it in 45 randomly chosen time-steps out of 50. Our goal is to test how such highly partial observations can be accumulated and retained over the long-term. The models were trained in these settings before performing validation. In Appendix C.1, we describe the architectures of SNP and the baseline NP for the 1D regression setting.

We present our quantitative results in Fig. 4. We report the target NLL on a held-out set of 1600 episodes computed by sampling the latents conditioned on the context as in Kim et al. (2019). In task (a), in the absence of context for $t \in [11, 20]$ we expect the transition noise to accumulate for any model since the underlying true dynamics are also noisy. We note that in contrast to NP, SNP shows less degradation in prediction accuracy. In task (b) and (c) as well, the proposed SNP outperforms the NP baseline. In fact, SNP's accuracy improves with accumulating context while NP's accuracy deteriorates with time. This is particularly interesting because NP is allowed to access the past context directly whereas SNP is not. This demonstrates a more effective transfer of past knowledge in contrast to the baseline. More qualitative results are provided in Appendix A.1 (Fig. 9). PD was not particularly crucial for training success on the 1D regression tasks (see Fig. 4). Fig. 2 compares the sampled functions.

### 5.2 2D and 3D Dynamic Scene Inference

We subject our model to the following 2D and 3D visual scene environments. The *2D environments* consist of a white canvas having two *moving* objects. Objects are picked with a random shape and color which, to test stochastic transition, may randomly be changed once in any episode with a fixed rule e.g., red $\leftrightarrow$ magenta or blue $\leftrightarrow$ cyan. When two objects overlap, one covers the other based on a fixed rule (See Appendix D.2). Given a 2D viewpoint, the agent can observe a $64 \times 64$-sized cropped portion of the canvas around it. The *3D environments* consist of movable object(s) inside a walled-enclosure. The camera is always placed on a circle facing the center of the arena. Based on the camera's angular position $u$, the query viewpoint is a vector $(\cos u, \sin u, u)$. We test the following two 3D environments: *a*) *Color Cube Environment* contains a cube with different colors on each face. The cube moves or rotates at each time-step based on the translation actions (Left, Right, Up, Down) and the rotation actions (Anti-clockwise, Clockwise) *b*) *Multi-Object Environment:* The arena

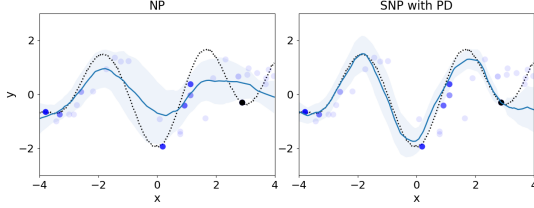

**Figure 2:** Sample prediction in 1D regression task (c) at $t = 33$. *Blue dots:* Past context. *Black dots:* Current context. *Black dotted line:* True function. *Blue line:* Prediction. *Blue shaded region:* Prediction uncertainty.

| Dataset | Regime | $T$ | GQN | TGQN | |
|---|---|---|---|---|---|
| | | | | no PD | PD |
| Color Shapes | Predict | 20 | 5348 | 489 | 564 |
| Color Cube (*Det.*) | Predict | 10 | 380 | 221 | 226 |
| Multi-Object (*Det.*) | Predict | 10 | 844 | 346 | 357 |
| Color Shapes | Track | 20 | 5285 | 482 | 513 |
| Color Cube (*Jit.*) | Track | 20 | 783 | 153 | 156 |
| Multi-Object (*Jit.*) | Track | 20 | 1777 | 450 | 475 |

**Table 1:** Negative $\log p(Y|X, C)$ estimated using importance-sampling from posterior with $K = 40$.

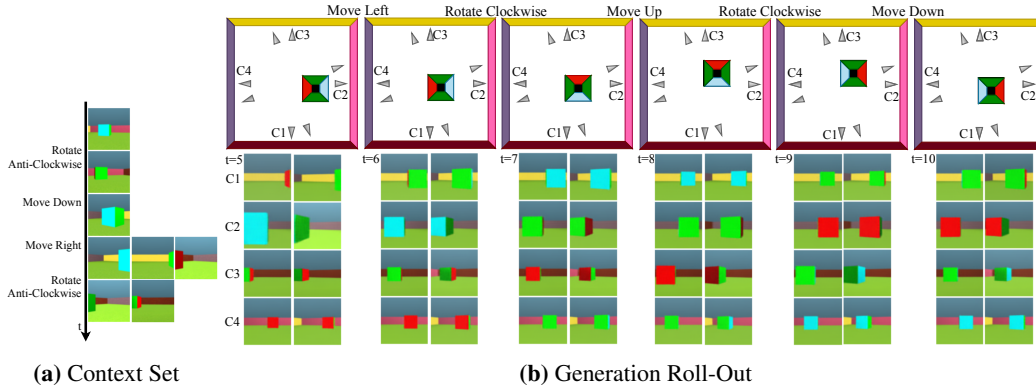

**(a)** Context Set

**(b)** Generation Roll-Out

**Figure 3:** TGQN demonstration in Color-Cube Environment. *Left:* The contexts and actions provided in $t < 5$. *Top Right:* Scene maps showing the queried camera locations and the true cube and the wall colors. *Bottom Right:* TGQN predictions in $5 \leq t \leq 10$.

contains a randomly colored sphere, a cylinder and a cube with translation actions given to them (see Appendix D.3). The action at each time-step is chosen uniformly. The 3D datasets have two versions: *deterministic* and *jittery*. In the former, each action has a deterministic effect on the objects. In the jittery version, a small Gaussian jitter is added to the object motion after the action is executed. The purpose of these two versions is described next.

**Context Regimes.** We explore two kinds of context regimes: *prediction* and *tracking*. In the *prediction* regime, we evaluate the model's ability to predict future time-steps without any assistance from the context. So we provide up to 4 observations in each of the first 5 time-steps and let the model predict the remaining time-steps (guided only by the actions in the 3D tasks). We also predict beyond the training sequence length ($T = 10$) to test the generalization capability. This regime is used with the 2D and the deterministic 3D datasets. In the *tracking* regime, we seek to demonstrate how the model can transfer past knowledge while also meta-learning the process from the partial observations of the current time-step. We, therefore, provide only up to 2 observations at every time-step of the roll-out of length $T = 20$. We test this regime with the 2D and the jittery 3D datasets since, in these settings, the model would keep finding new knowledge in every observation.

**Baseline and Performance Metrics.** We compare TGQN to GQN as baseline. Since GQN's original design does not consume actions, we concatenate the camera viewpoint and the RNN encoding of the action sequence up to that time-step to form the GQN query. In the action-less environments, the query is the camera viewpoint concatenated with the normalized $t$ (see Appendix C.3). We report the NLL of the entire roll out $-\log P(Y|X, C)$ estimated using 40 samples of $Z$ from $Q(Z|C, D)$. To report the time-step wise generation quality, we compute the pixel MSE per target image averaged over 40 generated samples using the prior $P(Z|C)$.

**Quantitative Analysis.** In Table 1 and Fig. 4, we compare TGQN trained with posterior dropout (PD) versus GQN and versus TGQN trained without PD. TGQN outperforms GQN in all environments in both NLL and pixel MSE. In terms of image generation quality in the *prediction* regime, the

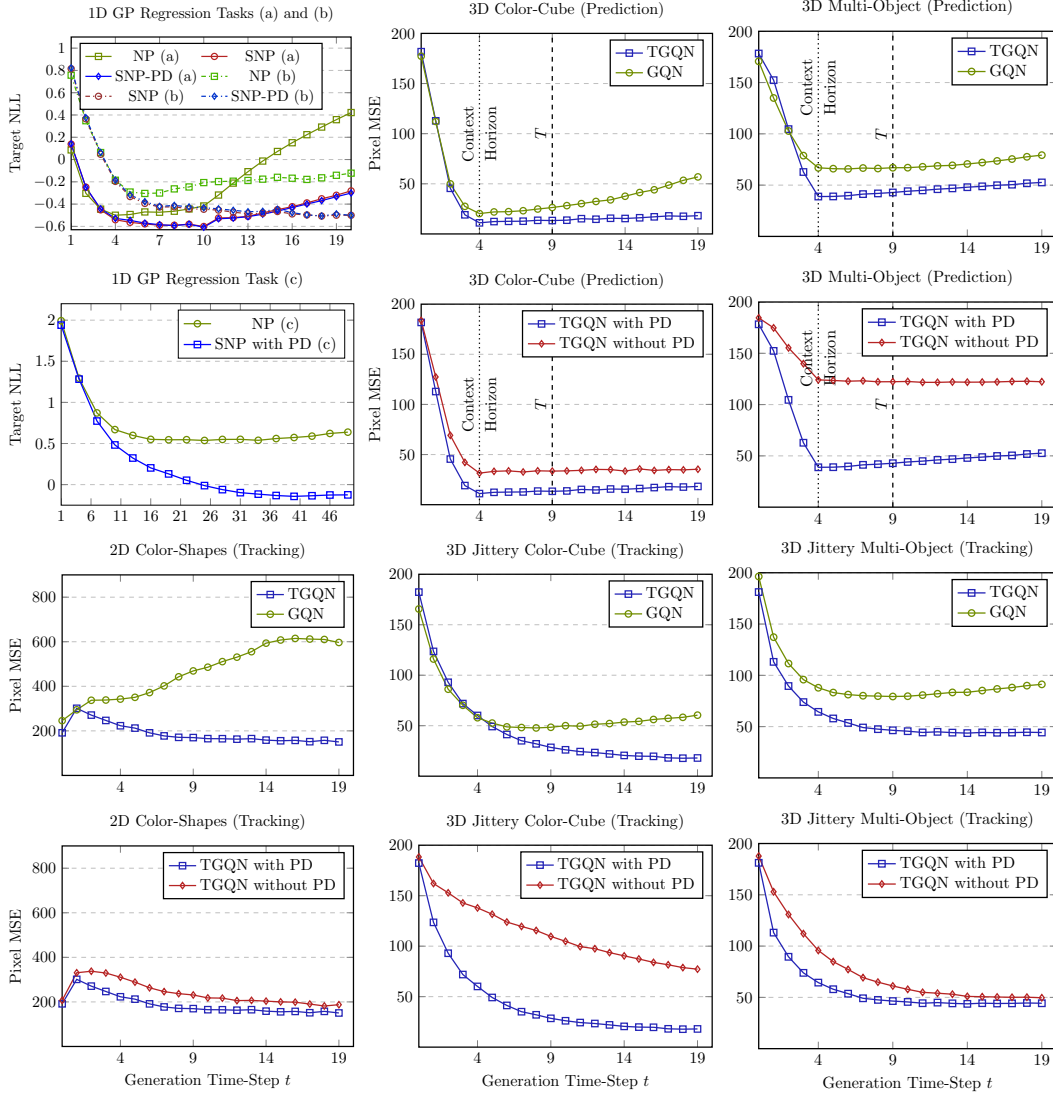

**Figure 4:** Comparison of generations of SNP with NP or GQN and comparison between SNP with and without posterior-dropout (PD). The latents are rolled-out from the prior conditioned on the context. For 1D regression, we report target NLL. For 2D and 3D settings, we report pixel MSE per image generated at each time-step.

pixel MSE gap is sustained even beyond the training horizon. In *tracking* regime, TGQN with PD converges in the fewest time-steps of observing the contexts. While TGQN continually improves by observing contexts over time, GQN's performance starts to deteriorate after a certain point. This is interesting since GQN can directly access all the past observations. This demonstrates TGQN's better temporal modeling and transfer of past knowledge. In general, the use of PD improves generation quality in all the explored cases. However, we note that the NLL of TGQN with PD is slightly higher than TGQN without PD. This is reasonable because TGQN with PD does not ignore $C_t$ when the past scene modeling in $z_{<t}$ is incorrect. This means that the model must carry extra modeling power to temporarily model the incorrect scene until more observations are available and then remodel the correct scene latent. This explains the tendency towards a slightly higher NLL.

**Qualitative Analysis.** In Fig. 3, we show a demonstration of TGQN's predictions for the Color Cube task. In Fig. 5, we qualitatively show the TGQN generations compared against the true images and the GQN generations. We infer the following from the figure. *a)* The dynamics modeled using $p_\theta(z_t|z_{<t}, C_t)$, can be used to sample long possible futures. This differentiates our modeling from the baselines where a single latent $z$ must compress all the indefinite future possibilities. In the 2D

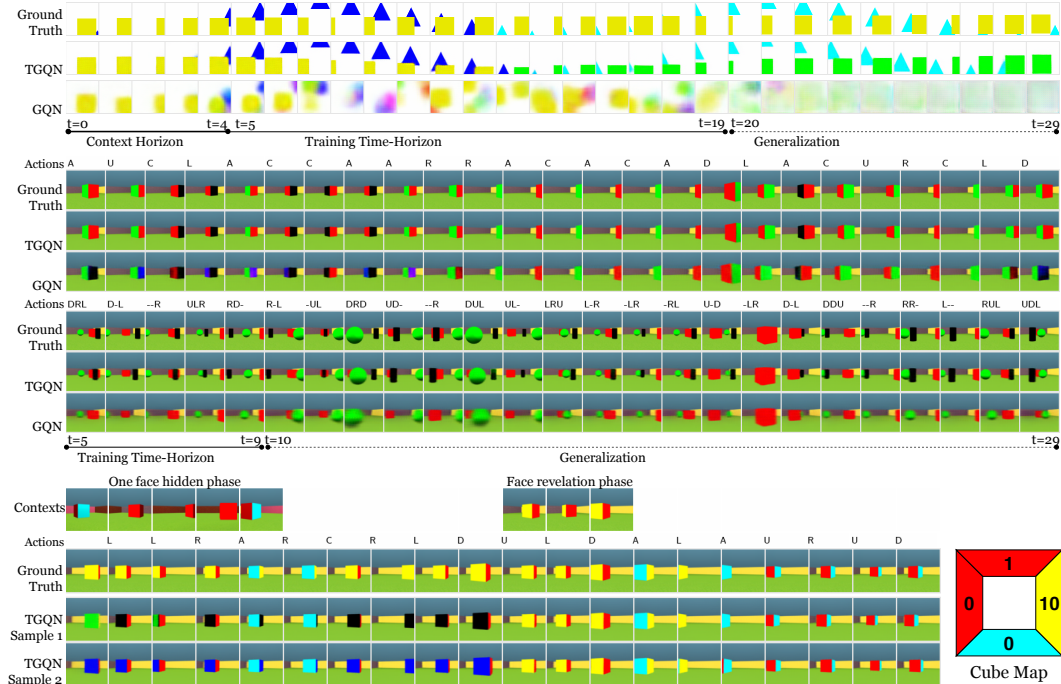

**Figure 5:** Qualitative comparison of TGQN with GQN (more in Appendix A.1 and A.2). *Top:* Prediction and generalization in 2D and deterministic 3D tasks. *Bottom:* Uncertainty modeling and meta-transfer learning in 3D jittery color-cube data set. The cube map shows the true face colors and the time-step at which it is revealed.

task, TGQN keeps generating plausible shape, motion and color changes. GQN fails here because the sampled $z$ does not contain information beyond $t = 20$, its training sequence-length. *b*) In the Color Cube and the Multi-Object tasks, we observe that TGQN keeps executing the correct object transitions. In contrast, GQN is susceptible to forgetting the face colors in longer-term generations. Although GQN can generate object positions correctly, this can be credited to the RNN that encodes the action sequence into the query. (Note that this RNN action-encoding is what we additionally endow to the vanilla GQN to make a strong baseline.) However, since this RNN is deterministic, this modeling would fail to capture stochasticity in the transitions. *c*) GQN models the whole roll-out in a single latent. It is therefore limited in its capacity in modeling finer details of the image. We see this through the poorer reconstruction and generation quality in the 3D tasks. *d*) TGQN can model uncertainty and perform *meta-transfer learning*. We test this in the jittery color-cube task by avoiding revealing the yellow face in the early context and then revealing it at a later time-step. When the yellow face is unseen, TGQN samples a face color from the true distribution. Upon seeing the face, it updates its belief and makes the correct color while still remembering the face colors seen earlier.

## 6   Conclusion

We introduced SNP, a generic modeling framework for meta-learning temporally-evolving stochastic processes. We showed that this allows for richer scene representations evidenced by the improved generation quality that can generalize to longer time-horizons in contrast to NP and GQN while also performing meta-transfer learning. We resolved the problem of transition collapse in training SNP using posterior dropout. This work leaves multiple avenues for improvement. NPs are susceptible to under-fitting (Kim et al., 2019) and it may also be the case with SNP. It would be interesting to see how the efficiency on the number of observations needed to meta-learn new information could be improved. It would also be interesting to see if an SNP-augmented RL agent can perform better in meta-RL settings than the one without.

**Acknowledgments**

This work was supported by Electronics and Telecommunications Research Institute (ETRI) grant funded by the Korean government. [19ZH1100, Distributed Intelligence Core Technology of Hyper-Connected Space]. SA thanks to Kakao Brain, Center for Super Intelligence (CSI), and Element AI for their support. JY thanks to Kakao Brain and SAP for their support.

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
