[Supplementary Material]

# Appendix A  Additional Demonstrations of SNP

In this section, we show additional qualitative demonstrations of SNP and comparisons against NP, GQN and the ground truth.

## A.1  Uncertainty modeling and meta-transfer learning in SNP

**Figure 6:** Goal of this demonstration is to show uncertainty modeling and meta-transfer learning in TGQN and GQN in the jittery color-cube environment. We provide contexts in two phases. In the early phase ($t = 0$ to 4), we show one observation per time-step while avoiding revealing a particular face. In the late phase ($t = 10$ to 12), we reveal that face. On the left, we show true face colors of the cube with the numbers showing the time-step at which it is first revealed. We generate two samples of rollouts of TGQN and GQN each. We make the following observations. *i)* In time-steps 5 through 9, we observe that TGQN can model uncertainty when the face colors are unseen and samples a color from the true palette. *ii)* In time-steps 10 through 12, we note that the face revelation updates the color of the previously unseen face. Since the cube dynamics are jittery, we also note that this context re-synchronises the cube position. Furthermore, we observe that TGQN transfers its knowledge of previously seen faces and combines it with the newly revealed face, thus performing meta-transfer learning. This new knowledge is maintained in the predictions made henceforth. *iii)* Overall, GQN produces blurred generations with inconsistency in the colors of the unseen faces.

**Figure 7:** Goal of this demonstration is to show tracking ability of TGQN versus GQN in the jittery multi-object environment. We provide contexts in two phases. In the early phase ($t = 0$ to $4$), we show one context observation at each time-step. Then we let the model predict the next 15 time-steps so that the predictions diverge from the true scene due to the jitter in the dataset. In the synchronisation phase ($t = 20$ to $22$), we show one observation per time-step. In this demonstration, we show two samples of the rollout from TGQN and GQN each from time-steps 5 through 29. We make the following observations. *i)* In time-steps 5 through 19, TGQN shows that it appropriately models the transition stochasticity as different samples produce different object positions at $t = 19$. *ii)* At $t = 20$, we see that the context re-synchronises the object positions with the true positions. *iii)* Overall, GQN produces blurred generations and is not able to model the cylinders.

**Figure 8:** Goal of this demonstration is to show tracking ability of TGQN versus GQN in the 2D color-shapes dataset. We provide contexts in two phases. In the early phase ($t = 0$ to $4$), we show one context observation at each time-step. Then we let the model predict the next 5 time-steps so that the predictions diverge from the true scene due to the random color-change in the dataset. In the synchronisation phase ($t = 10$ to $12$), we show one observation per time-step. In this demonstration, we show two samples of the rollout from TGQN and GQN each from time-steps 0 through 19. We make the following observations. *i)* In time-steps 5 through 9, TGQN shows that it appropriately models the transition stochasticity as different samples produce different object positions and color-changes. *ii)* At $t = 10$, we see that the context re-synchronises the object positions and colors with the true ones. *iii)* Overall, GQN produces blurred generations.

**(a)** Episode 1        **(b)** Episode 2

**Figure 9:** 1D regression qualitative samples for task (c). Each row corresponds to one time-step. Due to space limitations, every $5^{\text{th}}$ time-step is shown here instead of every time-step up to 45.

## A.2 Prediction in SNP

In this section, we demonstrate the predictions using SNP.

**Figure 10:** The goal of this demonstration is to show predictions from $t = 5$ through 29 using the context shown only in the early time-steps $t < 5$. Each row shows views from cameras positioned at angles labelled on the right. We compare TGQN with GQN and the ground truth. We observe that TGQN makes clear predictions even beyond the training sequence length $T = 10$. In contrast, GQN's generations are blurred with susceptibility to forgetting face colors.

**Figure 11:** The goal of this demonstration is to show predictions from $t = 5$ through 29 using the context shown only in the early time-steps $t < 5$. Each row shows views from cameras positioned at angles labelled on the right. We compare TGQN with GQN and the ground truth. We observe that TGQN makes clear predictions even beyond the training sequence length $T = 10$. In contrast, GQN's generations are blurred and it also cannot model the finer details like the cylinder.

**Figure 12:** The goal of this demonstration is to show predictions from $t = 0$ through 29 using the context shown only in the early time-steps $t < 5$. We compare TGQN with GQN and the ground truth. The labels on the right show where the queried patch is located e.g., *SW* indicates that the query patch is located at south-west corner of the canvas. We observe in time-steps $t \geq 5$, when the context is removed, that TGQN keeps making plausible predictions in accordance with the stochastic color change rules and motion dynamics. This continues even beyond the training sequence length $T = 20$. In contrast, GQN's predictions are blurred and cannot generalize beyond $T = 20$.

# Appendix B   ELBO Derivations

In this section, we derive the ELBO expressions that were introduced in the main text of the paper.

## B.1   SNP ELBO

In this sub-section we derive the ELBO mentioned in (6). We start with the objective of maximizing the log-likelihood of the targets given the queries and the contexts.

$$\log P(Y|X,C)$$

$$= \log \mathbb{E}_{Q_\phi(Z|\mathcal{V})} \frac{P(Y,Z|X,C)}{Q_\phi(Z|\mathcal{V})}$$

$$= \log \mathbb{E}_{Q_\phi(Z|\mathcal{V})} \frac{\prod_{t=1}^T P_\theta(Y_t|X_t,z_t)P_\theta(z_t|z_{<t},C_t)}{\prod_{t=1}^T Q_\phi(z_t|z_{<t},\mathcal{V})}$$

$$\geq \mathbb{E}_{Q_\phi(Z|\mathcal{V})} \left[ \log \frac{\prod_{t=1}^T P_\theta(Y_t|X_t,z_t)P_\theta(z_t|z_{<t},C_t)}{\prod_{t=1}^T Q_\phi(z_t|z_{<t},\mathcal{V})} \right]$$

$$= \mathbb{E}_{Q_\phi(Z|\mathcal{V})} \sum_{t=1}^T \left[ \log \frac{P_\theta(Y_t|X_t,z_t)P_\theta(z_t|z_{<t},C_t)}{Q_\phi(z_t|z_{<t},\mathcal{V})} \right]$$

$$= \mathbb{E}_{Q_\phi(Z|\mathcal{V})} \sum_{t=1}^T \left[ \log P_\theta(Y_t|X_t,z_t) + \log \frac{P_\theta(z_t|z_{<t},C_t)}{Q_\phi(z_t|z_{<t},\mathcal{V})} \right]$$

$$= \sum_{t=1}^T \mathbb{E}_{Q_\phi(Z|\mathcal{V})} \left[ \log P_\theta(Y_t|X_t,z_t) + \log \frac{P_\theta(z_t|z_{<t},C_t)}{Q_\phi(z_t|z_{<t},\mathcal{V})} \right]$$

$$= \sum_{t=1}^T \mathbb{E}_{Q_\phi(Z|\mathcal{V})} \left[ \log P_\theta(Y_t|X_t,z_t) - \log \frac{Q_\phi(z_t|z_{<t},\mathcal{V})}{P_\theta(z_t|z_{<t},C_t)} \right]$$

$$= \sum_{t=1}^T \mathbb{E}_{Q_\phi(z_t|\mathcal{V})} \left[ \log P_\theta(Y_t|X_t,z_t) \right] - \mathbb{E}_{Q_\phi(z_{\leq t}|\mathcal{V})} \log \frac{Q_\phi(z_t|z_{<t},\mathcal{V})}{P_\theta(z_t|z_{<t},C_t)}$$

$$= \sum_{t=1}^T \mathbb{E}_{Q_\phi(z_t|\mathcal{V})} \left[ \log P_\theta(Y_t|X_t,z_t) \right] - \mathbb{E}_{Q_\phi(z_{<t}|\mathcal{V})} \left[ \mathbb{KL}(Q_\phi(z_t|z_{<t},\mathcal{V}) \parallel P_\theta(z_t|z_{<t},C_t)) \right]$$

which gives us the expression in (6).

## B.2   Posterior Dropout ELBO

In this sub-section, we derive the ELBO with *posterior dropout* (8). As mentioned in Section 3.4, we choose a subset of time-steps $\mathcal{T}$ so that we use the prior distribution to sample the $z_t$ and posterior for the time-steps in $\tilde{\mathcal{T}}$. We start with the objective of maximizing the likelihood of the target images belonging to the time-steps in $\tilde{\mathcal{T}}$ and then proceed with the derivation as shown below.

$$\mathbb{E}_{\tilde{\mathcal{T}}} \log P_\theta(Y_{\tilde{\mathcal{T}}}|X,C)$$

$$= \mathbb{E}_{\tilde{\mathcal{T}}} \log \int \prod_{t\in\tilde{\mathcal{T}}} P_\theta(y_t|x_t,z_t) \prod_{t=1}^T P_\theta(z_t|z_{<t},C_t) dZ$$

$$= \mathbb{E}_{\tilde{\mathcal{T}}} \log \mathbb{E}_{Z\sim\tilde{Q}} \left[ \frac{\prod_{t\in\tilde{\mathcal{T}}} P_\theta(y_t|x_t,z_t) \prod_{t=1}^T P_\theta(z_t|z_{<t},C_t)}{\prod_{t\in\mathcal{T}} P_\theta(z_t|z_{<t},C_t) \prod_{t\in\tilde{\mathcal{T}}} Q_\phi(z_t|z_{<t},C,D)} \right]$$

$$\geq \mathbb{E}_{\tilde{\mathcal{T}}} \mathbb{E}_{Z\sim\tilde{Q}} \sum_{t\in\tilde{\mathcal{T}}} \left[ \log P_\theta(y_t|x_t,z_t) - \mathbb{KL}(Q_\phi(z_t|z_{<t},C,D) \parallel P_\theta(z_t|z_{<t},C_t)) \right] = \mathcal{L}_{\text{PD}}$$

which gives us the required expression in (8).

# Appendix C   Neural Networks

## C.1   Sequential Neural Processes and the baseline Neural Processes for 1D regression

SNP and the NP baseline have two encoders: *deterministic encoder* and *latent encoder*. This model does not consume actions. The deterministic encoder consists of a 6-layer MLP with ReLU (Nair & Hinton, 2010) activation. The latent encoder consists of a 3-layer MLP with ReLU followed a 2-layer MLP for computing sufficient statistics of the latent. This encoder acts as a prior when provided only with the context set, but also acts as the posterior when provided with the target set. We implement the state-space model using an LSTM with the default Tensorflow (Abadi et al., 2016) settings.

Since NP is not a temporal architecture, normalized time $t' = 0.25 + 0.5 \times (t/T)$ is appended to the original query $x$ to obtain $\tilde{x} = (x, t')$.

The dimension of the hidden units is 128. The learning rate and the batch size are 0.0001 and 16, respectively.

## C.2   Temporal Generative Query Networks

Here, we give the details of the implementation of the TGQN model geared towards generation of 3D scenes. Our implementation is fully convolutional i.e., all the latent states and deterministic states are 3 dimensional tensors.

**Generation**   Below, we outline the implementation of the generative model.

$$h_0 \leftarrow \text{learned parameter} \qquad \text{(Initialize deterministic state)} \qquad (9)$$
$$z_0 \leftarrow \text{learned parameter} \qquad \text{(Initial latent)} \qquad (10)$$
$$C_t \leftarrow \sum_{i=1}^{n_t} \text{RepNet}_\theta(x_i^t, y_i^t) \qquad \text{(Compute scene representation from observed context)} \quad (11)$$
$$a_t \leftarrow \text{action embedding} \qquad \text{(One-hot action embedding)} \qquad (12)$$
$$h_t \leftarrow \text{RNN}_\theta(h_{t-1}, z_{t-1}, a_{t-1}, C_t) \qquad \text{(Deterministic state transition)} \qquad (13)$$
$$z_t \sim \text{DRAW}_\theta(h_t, a_{t-1}, C_t) \qquad \text{(Sample } z_t \text{ using DRAW)} \qquad (14)$$
$$y_i^t \leftarrow \text{Renderer}_\gamma(x_i^t, z_t, h_t) \qquad \text{(Render the image)} \qquad (15)$$

More details about the implementation of $\text{DRAW}_\theta$, $\text{RepNet}_\theta$ and the $\text{Renderer}_\gamma$ are provided in following sections.

**Inference**   Next, we outline the inference procedure used for sampling all the latents $z_{1:T}$.

$$D_t \leftarrow \sum_{i=n_t+1}^{m_t} \text{RepNet}_\theta(x_i^t, y_i^t) \quad \text{(Compute scene representation from target observations)} \quad (16)$$
$$b_t \leftarrow \text{RNN}_\phi(b_{t+1}, C_t, D_t, a_t) \quad \text{(Encode all observations using a backward RNN)} \qquad (17)$$
$$z_t \sim \text{DRAW}_\phi(h_t, a_{t-1}, b_t) \qquad \text{(Sample } z_t \text{ using DRAW.)} \qquad (18)$$

Here, $h_0$ is the same as in (9). Next, we compute all the $h_t$'s and sample all the $z_t$'s by using $D_t + C_t$ instead of just $C_t$. The $h_t$'s for $t > 0$ are computed as in (13) using the generative network. All the $z_t$'s for $t > 0$ are drawn similar to (14) using $\text{DRAW}_\phi$. Note that $\text{DRAW}_\phi$ has access to the internal states of the generative $\text{DRAW}_\theta$ network. This has been omitted in (18) for brevity but is described in the following sections.

### C.2.1   Basic Building Blocks

1. **Representation Network:** The representation network takes an image-viewpoint pair and summarizes the scene as a 3D tensor. Multiple such representations are combined in an order-invariant fashion by summing or averaging. We use the Tower Network as described in Eslami et al. (2018).

$$D = \{(\mathbf{x}_1, \mathbf{y}_1), (\mathbf{x}_2, \mathbf{y}_2), \dots (\mathbf{x}_m, \mathbf{y}_m)\}$$
$$R_D = \frac{1}{m} \sum_{i=1}^{m} \text{RepNet}(\mathbf{x}_i, \mathbf{y}_i)$$

Here, $D$ is a set of image-viewpoint pairs and $R_D$ is its computed representation.

2. **Convolutional LSTM Cell:** A standard LSTM Cell where all fully-connected layers are substituted for convolutional layers.

$$(h_{i+1}, c_{i+1}) \longleftarrow \text{ConvLSTM}(\mathbf{x}_i, h_i, c_i)$$

where $h_i$ is the output of the cell and $c_i$ is the recurrent state of the ConvLSTM.

### C.2.2 Renderer $p(\mathbf{y}|\mathbf{z}, \mathbf{h}, \mathbf{x})$

The input to the renderer is the scene information contained in the latent $\mathbf{z}$ and deterministic state $\mathbf{h}$ along with the camera viewpoint $\mathbf{x}$. The output is the generated image $\mathbf{y}$. The renderer is deterministic and iterative where each iteration updates the image canvas as follows.

$$\mathbf{e}^{(i)} \leftarrow \text{encoder}(\mathbf{y}^{(i)})$$
$$(\mathbf{d}^{(i+1)}, \mathbf{c}^{(i+1)}) \leftarrow \text{ConvLSTM}(\mathbf{e}^{(i)}, \mathbf{d}^{(i)}, \mathbf{c}^{(i)}, \mathbf{x}, \mathbf{h}, \mathbf{z})$$
$$\mathbf{y}^{(i+1)} \leftarrow \mathbf{y}^{(i)} + \text{decoder}(\mathbf{d}^{(i+1)})$$

Here, $\mathbf{x}^{(i)}$ is the canvas at the $i^{\text{th}}$ iteration and the $\mathbf{d}^{(i)}$ and $\mathbf{c}^{(i)}$ are the hidden state and the cell state of the convolutional LSTM, respectively. The number of iterations is a model parameter taken as 6.

Next, we describe the details of the encoder and decoder used above.

1. **Encoder:** Details are shown in the Figure 13.

**Figure 13:** Encoder network has two convolutional layers. After each layer, ReLU non-linearity is applied.

2. **Decoder:** Details are shown in the Figure 14.

**Figure 14:** Encoder network has one convolutional layer and two transposed convolutional layers. After each layer except the last, ReLU non-linearity is applied.

### C.2.3 Updating the deterministic state $\mathbf{h}_t$

For any $t$, the deterministic state $\mathbf{h}_t$ summarizes all the previous latent states $\mathbf{z}_{<t}$. This deterministic state is updated using a convolutional LSTM. The update may be described as follows.

$$(\mathbf{h}_{t+1}, \mathbf{c}_{t+1}) \leftarrow \text{ConvLSTM}(\mathbf{z}_t, \mathbf{a}_t, \mathbf{h}_t, \mathbf{c}_t)$$

Here, $\mathbf{c}_t$ is the LSTM's internal cell state and $\mathbf{a}_t$ is the action received at time $t$.

### C.2.4 Sampling the latent $\mathbf{z}_t$ using $p(\mathbf{z}_t | \mathbf{h}_t, \mathbf{a}_t)$

The sampling of latents, like CGQN (Kumar et al., 2018), is done using a DRAW-like auto-regressive density. Assume that $a$) $\mathbf{h}$ is the deterministic state, $b$) $\mathbf{a}$ is the action provided, $c$) $C$ is the context encoding provided at the current time-step and $d$) $D$ is the target encoding provided at the current time-step.

**Generation**  This procedure is described in the following equations.

$$(\hat{h}_0^p, \hat{c}_0^p) \leftarrow \text{learned parameters} \qquad \text{(Initial RNN state for generation)} \qquad (19)$$

$$(\hat{h}_l^p, \hat{c}_l^p) \leftarrow \text{RNN}_\theta(z_t^{l-1}, \hat{h}_{l-1}^p, \hat{c}_{l-1}^p, \mathbf{h}, \mathbf{a}, C) \qquad \text{(Update rule for generative RNN)} \qquad (20)$$

$$(\mu^l, \sigma^l) \leftarrow \text{SufficientStatistics}_\theta(\hat{h}_l^p) \qquad \text{(See Fig. 15)} \qquad (21)$$

$$z^l \sim \mathcal{N}(\mu^l, \sigma^l) \qquad \text{(Sample the latent at current DRAW step)} \qquad (22)$$

**Figure 15:** Computing sufficient statistics from the RNN hidden state of the auto-regressive density.

**Inference**  The inference procedure performs a similar sampling of the $z^l$'s but while having access to the hidden state of the generative RNN computed in (20). This procedure is described in the following equations.

$$(\hat{h}_0^p, \hat{c}_0^p) \leftarrow \text{learned parameters} \qquad \text{(Initial RNN state for generation)} \qquad (23)$$

$$(\hat{h}_0^q, \hat{c}_0^q) \leftarrow \text{learned parameters} \qquad \text{(Initial RNN state for inference)} \qquad (24)$$

$$(\hat{h}_l^q, \hat{c}_l^q) \leftarrow \text{RNN}_\theta(z_t^{l-1}, \hat{h}_{l-1}^q, \hat{h}_{l-1}^p, \hat{c}_{l-1}^q, \mathbf{h}, \mathbf{a}, D) \qquad \text{(Update rule for inference RNN)} \qquad (25)$$

$$(\mu^l, \sigma^l) \leftarrow \text{SufficientStatistics}_\theta(\hat{h}_l^q) \qquad \text{(See Fig. 15)} \qquad (26)$$

$$z^l \sim \mathcal{N}(\mu^l, \sigma^l) \qquad \text{(Sample the latent at current DRAW step)} \qquad (27)$$

$$(\hat{h}_l^p, \hat{c}_l^p) \leftarrow \text{RNN}_\theta(z_t^{l-1}, \hat{h}_{l-1}^p, \hat{c}_{l-1}^p, \mathbf{h}, \mathbf{a}, C) \qquad \text{(Update rule for the generative RNN)} \qquad (28)$$

### C.2.5 Hyper-Parameters

In this sub-section, we describe the hyper-parameters used in our training.

| Parameter | 3D Tasks | 2D Tasks |
|---|---|---|
| Image Width/Height | 64 | 64 |
| Image Channels | 3 | 3 |
| Latent Width/Height | 16 | 16 |
| Renderer Image Encoding Depth | 128 | 128 |
| ConvLSTM Hidden State Depth | 128 | 128 |
| Context Representation Depth | 256 | 256 |
| SSM Transition State Depth | 108 | 108 |
| Training Batch-Size | 4 | 4 |
| Latent Depth per DRAW step | 4 | 4 |
| Action Input Embedding | One-hot | N/A |
| Number of DRAW steps | 6 | 6 |
| Learning Rate | $10^{-5}$ | $10^{-5}$ |
| Viewpoint Size | 3 | 2 |
| RGB Distribution | Gaussian | Gaussian |
| RGB $\sigma^2$ | 2.0 | 2.0 |
| Maximum context per time-step | 4 | 4 |

**Posterior Dropout**   requires that we randomly choose between using $P_\theta$ or $Q_\phi$. The choice was made using a Bernoulli coin-toss with probability 0.3 (for $Q_\phi$) at every time-step of each episode for each training iteration. Furthermore, the training of any task was first initiated without the posterior dropout ELBO i.e. with $\alpha = 0$. The posterior dropout ELBO was turned on, i.e. setting $\alpha = 1$, after the reconstruction loss using the SNP ELBO had saturated. This is done to avoid conflict in the training of the encoder network due to two competing reconstruction losses from the two ELBOs in the initial stages of the training.

### C.3 GQN Baseline

Here, we provide some salient details of our implementation the GQN baseline. *a*) In environments with actions, the query is a concatenation of the camera viewpoint and the RNN encoding of the action sequence up to that time-step. This RNN encoding has size 32. In action-less environments, $t$ as a normalized scalar concatenated to the camera viewpoint. *b*) We encode contexts (or targets) from multiple time-steps using sum-pooling as in original GQN. *c*) During generation, TGQN cannot observe contexts from future time-steps. So for fair comparison at generation time, we also provide GQN with an encoding of contexts only up to the time-step that we are interested in querying.

## Appendix D   Dataset Additional Details

### D.1   Gaussian Processes Dataset

In each episode of task (a) and (b), the hyper-parameters of length-scale $l \in [0.7, 1.2]$ and kernel-scale $\sigma \in [1.0, 1.6]$ are randomly drawn at $t = 0$. In the task (c), $l$ and $\sigma$ are drawn from ranges $[1.2, 1.9]$ and $[1.6, 3.1]$, respectively. Similarly, the linear dynamics $\Delta l \in [-0.03, 0.03]$ and $\Delta \sigma \in [-0.05, 0.05]$ are also drawn randomly at $t = 0$. To perform transitions, we execute $l + \Delta l$ and $\sigma + \Delta \sigma$ and add a small Gaussian noise at each time-step.

For task (a) and (b), the number of context and target are drawn randomly from $n \in [5, 50]$ and $m \in [0, 50 - n]$ whenever a non-empty context is being provided else $n = 0$ and $m \in [0, 50]$. For task (c), $n$ is 1 and $m$ is in $[0, 10 - n]$ whenever a non-empty context is being provided else $n = 0$ and $m \in [0, 10]$.

## D.2 2D Color Shapes Dataset

The canvas and object sizes are $130 \times 130$ and $38 \times 38$, respectively. Speed of each object is 13 pixels per time-step and the initial direction is randomly chosen. The bouncing behaviour is modeled the same way as in the moving MNIST dataset (Srivastava et al., 2015). Shapes can be triangles, squares or circles and their colors can be red, magenta, blue, cyan, green or yellow. Here, we provide the fixed rule that we use to decide which object covers the other in case of an overlap.

- Green or yellow cover red and magenta.
- Red or magenta cover blue and cyan.
- Magenta covers red.
- Cyan covers blue.
- Yellow covers green.

In this task, we pick the patch location (viewpoint) uniformly on the canvas. In the prediction regime, in each of the first 5 time-steps, we randomly decide the context set size $n$ uniformly in range $[1, 5]$ and the target size $m$ is then taken as the number of remaining observations $20 - n$. In the tracking regime, $n$ at each time-step is chosen in the range $[0, 2]$ the remaining observations are used as the target.

## D.3 3D Environment Details

We used the MuJoCo framework and the OpenAI Gym toolkit (Mordatch et al.; Brockman et al., 2016) to generate the 3D datasets. For training, we created 50,000 episodes where each episode contains 10 time-steps and each time-step contains 20 images. Therefore, the training is performed on 10 million images. For testing and evaluation, datasets containing 10,000 episodes with 30 time-steps each were separately generated.

Actions at each time-step are uniformly randomly picked. If an action leads the object outside the arena, the action is re-picked until it doesn't. At each time $t$, we take 20 random camera angles in $[0, 2\pi)$ and we use a part of it as context and leave the remaining as target. In the prediction regime, in each of the first 5 time-steps, we randomly decide the context set sizes uniformly in range $[1, 5]$. In the tracking regime, $n$ at each time-step is chosen in the range $[0, 2]$ the remaining observations are used as the target.