[Reviews · NeurIPS 2019]

Reviewer 1



The paper presents an extension to the Neural Processes (NP) model for dynamical systems. In more abstract terms, whereas the standard NP is a model of functions which can be defined on arbitrary domains X, in the proposed model the function domain is assumed to contain a dimension which evolves with some sequential model. The authors demonstrate the model on GP functions with dynamic kernels, and on images of dynamic 3D scenes. The paper presents an interesting idea which can be very useful in practice when modeling data that contains dynamic elements. The presentation of the model and background is written really well. The main issue in the paper is the lack of details in the experimental section, and in regards to the what the authors call the "transition collapse". Another issue is that the paper does not discuss the main drawback of their approach, namely that conditioning the model on a sequential domain rather than an arbitrary domain, restricts the predictions of the model to be sequential too. In contrast to the standard NP the proposed model cannot make "jumpy" predictions far in the future, or prediction of the past given the future. This is OK but needs to be clearly mentioned. The missing details are: 1. What is the transition model for the kernel in the regression task? 2. How was the NLL in the regression task computed? On held out sets? how many examples were used? 3. How was time used in the regression baseline? Was it part of the context encoding? Without this detail I'm not convinced the comparison is fair. 4. What is the motion model in the 2D images? What is the action policy in the 3D model? 5. The author mention the observations in the 2D data are 64x64 pixels. What is the size of the full image? 6. How were the actions and time encoded in the baseline's context (not only the query)? Appendix c mentions something about a backward RNN but that was not clear to me. 7. The description of the "Performance Metric" is inconsistent. NLL is not equivalent to the MSE of the mean. NLL is intractable but ELBO should be a decent proxy (roughly MSE/var + KL). Figure 3 says NLL and line 279-286 describe recall-MSE - which one is used? The discussion of the 'transition collapse" is interesting but it's hard to learn something without more empirical demonstrations and analysis: 1. Why is the PD loss annealed in this linear way? I would think a more natural way to implement annealing would be y varying T-tilde. 2. What was the alpha used? 3. What is the crucial difference between the two 3D datasets that makes PD vs. no-PD behave differently? 4. Can a toy example be constructed that shows exactly when "transition collapse" occurs and when dropout becomes usefull? I realize that it is hard to fit a full analysis of this in an 8 pages paper, so perhaps it would be better to treat this as a training detail (which should still be mentioned and discussed briefly. maybe having some analysis in the appendix), and add more space to describe the sequential vs. baselines experiments. Minor comments: Line 53: This is the first time CGQN is mentioned. Perhaps the place to give the full name and citation. Line 68-69: This sentence seems to be reversed. Shouldn't this be "condition the observation model on the context"? Equation 6: If I understand correctly the Q's of the expectations should be the same as in equation 5. i.e. Q_\phi (z_t | z_{

Reviewer 2



Summary: The authors propose a new model called the Sequential Neural Process that models a sequence of stochastic processes that show temporal dependency. This is done by formulating a Neural Process for each time step (with shared parameters) and modelling dependence between the latent variables of the NPs with a state-space model. Similar to NPs, variational inference with the reparameterisation trick is used for learning, but with a sequential variational distribution. The model is applied to a sequence of GPs with kernel parameters evolving through time as well as a 2D and 3D shape environment with moving objects, and compared empirically to CGQNs. The paper tackles a novel problem of modelling a sequence of stochastic processes and is technically sound. The writing is clear for the most part, and the results are well-organised and strong. The work addresses the difficult task of modelling a dynamic scene (i.e. with moving objects) from images of varying viewpoints, and shows results that are significant improvements from the baseline CGQN. The ablation study for the posterior dropout learning is also convincing. Overall, this is a very strong submission. Here below are some suggestions for the paper that may help to improve it. 1. I think it should be made even clearer that the task of SNPs is to model a changing sequence of stochastic processes by emphasising that SNP does NOT model a stochastic process that is a sequence (time series). The latter, being a more common problem than the former, might be what a reader might expect when reading the title and abstract, and may confuse the reader. Having said that, it would be interesting to see experimentally if the latter can be achieved in the SNP framework, e.g. by setting C_t, D_t as singleton data points (x_t,y_t) where x_t=t. One issue is that for extrapolation, one would have to deal with empty C_t for t>T. Have you explored this avenue? 2. In Section 2 when you introduce the Neural Process, you introduce the encoder by saying it is used to define the conditional prior p(z|C) in line 70-71. In this case, it would be helpful to clarify what the “intractable posterior” that you mention in line 75 actually is. It is proportional to p(z|C)p(Y|X,z), and you would usually denote this as p(z|C,D), but you can’t in this case because it’s not equal to a different conditional prior p(z|CUD) that you get when feeding in CUD to the encoder. And also in Equation (2), the variational distribution Q_{phi}(z|C,D) is in fact just the conditional prior p_theta(z|CUD) in practice. I think it can be misleading if you use phi for Q and theta for p, since phi=theta in this case. 3. I think it would be better to move Figure 1 to Section 3.3 when you describe TGQN, and for Section 3.1 add instead a figure with a graphical model of SNP. It can be confusing to put the figure for TGQN next to the description of SNP because e.g. you haven’t defined a_{t-1} that appears in the figure. 4. The example that you use to show SNPs are a meta-transfer learning method in lines 141-146 is nice, but since this example isn’t what you apply SNPs to in the experiments, I think it would be better to replace it with the example of the 3D environment with moving objects. This example of games against enemies can be moved to the discussion as an example of a more realistic scenario for a potential application of SNPs. 5. In line 177 what is meant by the sum of C_t and D_t? These two are both sets as far as I understand. 6. In line 183-187, you give an explanation of why transition collapse happens. Could this be made clearer by saying something along the lines of “the information about C_t is already present in the sampled z_{

Reviewer 3



This is an interesting paper, combining the features of NPs with dynamic latent-variable modelling through RSSMs. [Originality] The precise combination of NPs with RSSMs (or similar) models appears to be novel. As is the application to a dynamic 3D data domain. [Quality] The formulation and algorithmic model appears to be sound. I do have one particular concern to do with the characterisation of the posterior dropout elbo---while its use for TGQNs might be apposite, the idea of sampling from the prior to prevent the posterior from overfitting is not new, especially for ELBOs. It is usually referred to as defensive importance sampling [1]. Extending this temporally is relatively trivial given that the application is conditionally independent given the (presumably global) choice of which time steps to apply it to. Also, while the model allows for change of query viewpoint, within a temporal sequence, at least going by the examples shown in the experiments, there does not appear to be change in viewpoint within a sequence. Is this by design? To what extent does the TGQN learn a general scene representation that can queried in the spirit of GQNs? At test time, for a fixed scene setting (fixed context C that is) do new query viewpoints produce consistent scenes as GQN purports to do? [Clarity] The paper is well written and well organised. The experiments we also well structured. [Significance] I do believe the results shown here, along with the algorithmic contributions will help in pushing the state of the art in learning dynamic scene representations forward. I hope that source code for the model and data will be made publicly available for the community to build on. [1] Weighted Average Importance Sampling and Defensive Mixture Distributions, Hesterberg, Technometrics 1995 **Update** I have read the authors' response and am happy with the comments. Having perhaps one example of non-fixed viewpoints would definitely help showcase the features of the proposed method better. And I'm glad the authors are committed to releasing source code for this work. The additional results presented to analyse posterior collapse also helped answer some questions I subsequently had, thanks to the thorough work done by the other reviewers.

[Author Response · NeurIPS 2019]

Thank you for the positive, constructive and in-depth reviews. We found the suggestions and comments to be very
helpful. Below, we summarize the main questions and comments raised by each reviewer and provide responses.

**[R1]** **Drawback of SNP.** We agree. We will add a discussion on this. **Transition model in the regression task.** In
Appendix D.1, we describe how length-scale $l$, kernel-scale $\sigma$, $\Delta l$, and $\Delta \sigma$ are chosen. To perform transition, we
execute $l + \Delta l$ and $\sigma + \Delta \sigma$ and add a small Gaussian noise. **NLL in the regression task** is estimated by MC sampling,
the same way as used in the Attentive Neural Processes (ANP) paper. We tested it on a held-out set of 1600 examples.
**Time in regression task.** Normalized time $t' = 0.25 + 0.5 \times (t/T)$ is appended to the original query $x$ to obtain
$\tilde{x} = (x, t')$ and used an MLP$(\tilde{x}, y)$ to encode the query together with the target $y$ for context encoding. **Motion model**
**and canvas size in the 2D task.** Shapes start at random positions on a *96×96-sized canvas* with a speed of 13 pixel
per time-step towards a randomly chosen direction. The bouncing behaviour is modeled the same way as in the moving
MNIST dataset. **Action in 3D tasks** is uniformly randomly picked. If an action leads the object outside the arena,
the action is re-picked until it doesn't. **How action and time is encoded in GQN baseline in 3D tasks.** We use a
*forward*-RNN to encode context and actions for generation using $r_t = \text{RNN}(r_{t-1}, C_t, a_t)$. For inference/training, a
backward-RNN similar to this is used to encode actions, context and targets of the *entire* episode. At $t$, action sequence
is encoded as $\tilde{a}_t$ by a forward-RNN as $\tilde{a}_t = \text{RNN}(\tilde{a}_{t-1}, a_t)$. Query to GQN at time $t$ is the concatenation $(x, \tilde{a}_t)$.
Deploying an RNN encoding of the action sequence, we believe this is somewhat a stronger baseline than the vanilla
GQN. **Performance Metric.** We thank for pointing out this. There was some confusion. What we actually used is
sample-based NLL estimation. We found our argument connecting MSE to NLL needs a fix. The recall-MSE should
be recall-NLL. The **linear PD loss annealing** was simply our initial trial that we found to work well empirically. We
agree that it is worth to try your suggestion of controlling $\tilde{T}$ instead of $\alpha$. **PD-$\alpha$ annealing.** $\alpha = 0$ in early training and
set to 1 after reconstruction loss saturates. Using probability 0.2-0.5 of picking posterior transition in $\tilde{\mathcal{T}}$ worked well
in practice. **Why PD and no-PD behave differently for the two 3D tasks.** For now, we hypothesize that using PD
could be more effective when the task is more complex because reducing the gap between posterior and prior without
PD could be easier for simple tasks. For the 3D multi-object case, because the latents need to model the dynamics of
multiple objects, the information gap between $z_{<t}$ and $C_t$ could be larger than that of single-object case, and this could
make using PD more effective. In the table below, we measured two $\mathbb{KL}$s. As shown in the first row $\mathbb{KL}$, there is not
much difference between using PD and not using it because $z_{<t}^{\text{posterior}}$ contains pretty abundant information. But for the
second row $\mathbb{KL}$, we see that the gain by using PD becomes clearer as the task becomes more complex in the order of
Multi-Object > Color-Cube > Color-Shapes. We agree that we need more investigation to understand PD better, it will
be helpful to have a **toy task to analyze posterior collapse**. We hope to include this in the camera-ready version.

| Task | Multi-Object | | Color-Cube | | Color-Shapes | |
|---|---|---|---|---|---|---|
| Loss Type | No PD | PD | No PD | PD | No PD | PD |
| $\mathbb{KL}(q(z_t\|z_{<t}^{\text{posterior}}, C_t, D_t) \,\|\, p(z_t\|z_{<t}^{\text{posterior}}, C_t))$ | 4.78 | 3.18 | 1.86 | 0.83 | 0.73 | 0.56 |
| $\mathbb{KL}(q(z_t\|z_{<t}^{\text{prior}}, C_t, D_t) \,\|\, p(z_t\|z_{<t}^{\text{prior}}, C_t))$ | 57.45 | 3.49 | 3.51 | 1.06 | 1.05 | 0.67 |

**[R2]** We thank for the positive and insightful review. We treasure all the points that would make our paper clearer
and more precise. We agree on all of them. To judiciously use space, we address the remaining comments below. **Do**
**we explore empty $C_t$ for $t > T$?** Yes, for 2D and 3D tasks, we show context only up to $t = 5$ and we demonstrate
the temporal generalization up to $t = 20$ or 30. **Posterior notation.** We thank for pointing out this. We followed the
argument and we will make it clearer in the camera-ready. $P_\theta \equiv Q_\phi$? We will clarify that "in practice $\phi = \theta$". **Sum of**
$C_t$ **and** $D_t$**.** We meant the sum of the respective vector encodings but we agree it is more apt to say "$C_t \cup D_t$". **PD's $\alpha$**
**sensitivity.** We agree this needs more study. We responded with some details in line 20 above. **Details about the 1D**
**task.** In sub-tasks (a) and (b) we train the model under those settings before validating. For regression tasks, dynamics
are actionless. Our training time-horizon was $T = 20$ for tasks (a) and (b) and $T = 50$ for task (c). **Choosing $C_t, D_t$**
**for TGQN.** At each time $t$, we take 20 random camera angles in $[0, 2\pi)$ and we use a part of it as context and leave
the remaining as target. In each of the first 5 time-steps, we randomly decide the context set sizes uniformly in ranges
$[1, 3]$ and $[1, 5]$ for 2D and 3D tasks, respectively. For 2D task, we pick the patch location (viewpoint) uniformly on
the canvas. **Uncertainty demonstration.** Due to limited space, we initially could not fit it in the main body but we
would find a way to emphasize it more. We will also properly emphasize the fact that "**SNP's main motivation is *not***
**to model a stochastic process that is a sequence.**"

**[R3]** Thanks for the positive review and the reference to *defensive importance sampling*. It helps build a better
motivation for the PD loss. **Can we query any viewpoint (in query space) in the spirit of GQN?** Yes, the regularly
spaced viewpoints in diagrams are only for illustration to ease the reader in following object motion. **In face-**
**uncertainty, does consistency hold per scene?** Yes, we will add illustration for this in uncertainty demonstration.
However, more study is needed on consistency across time-steps since each $t$ has its own latent. **Code.** We will make it
available.

[Meta-Review · NeurIPS 2019]

I thank the authors for their submission. The paper presents an extension to the Neural Processes model for dynamical systems. I strongly encourage the authors to take into account the reviewers' comments and concerns for the final manuscript.